# Tactile Memory with Soft Robot: Tactile Retrieval-based Contact-rich Manipulation with a Soft Wrist

**Tatsuya Kamijo, Mai Nishimura, Cristian C. Beltran-Hernandez, and Masashi Hamaya**

OMRON SINIC X Corporation

`masashi.hamaya@sinicx.com`

**Abstract:** Tactile memory—the ability to store and retrieve touch-based experiences—is critical for humans to perform contact-rich and fine manipulation tasks like key insertion, even under uncertainties. Replicating this capability in robots remains challenging due to underdeveloped spatiotemporal representations for tactile signals. This study introduces TaMeSo-bot (Tactile Memory with Soft Robot), a robotic system that combines physical softness for safe contact with retrieval-based manipulation. Inspired by neurophysiological findings on tactile memory, TaMeSo-bot introduces a transformer-based method that processes multi-modal sequences—including tactile, force-torque, and proprioceptive signals—while modeling the spatial relationships across distributed taxel sensors. Leveraging a masked token prediction technique, our system autonomously extracts task-relevant features without manual subtask segmentation. We validate our approach on peg-in-hole tasks in both offline and real-robot experiments. Results show improved action position retrieval accuracy (34% over baseline) and performance with 77.5% and 57.5% success rates under seen and unseen conditions (peg and hole pose uncertainty and different diameter pegs), respectively.

**Keywords:** Soft Robotic Learning, Tactile Sensors, Contact-rich Manipulation

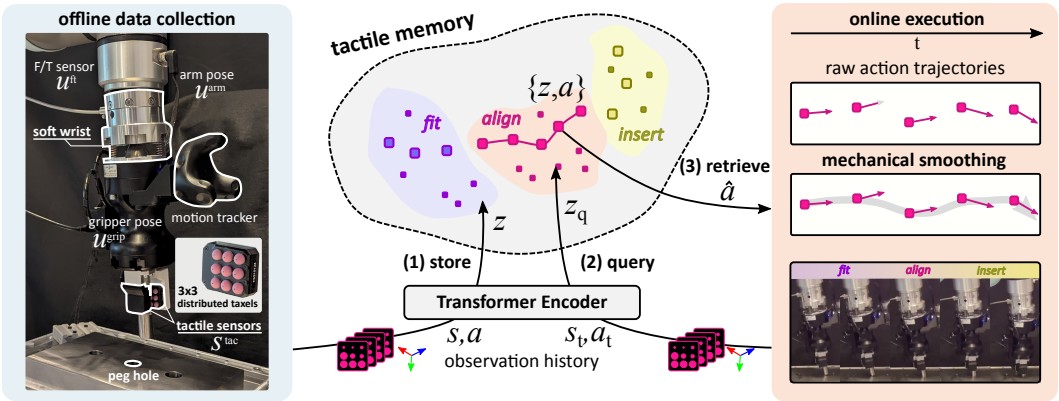

Figure 1: Overview of the **TaMeSo-bot** system. The tactile memory system stores encoded tactile demonstrations and retrieves relevant actions by matching current sensory inputs to similar past experiences, enabling robust contact-rich manipulation. The retrieved actions are then executed via mechanical smoothing through the soft wrist's physical compliance, which naturally filters discontinuities for safe and stable manipulation.

8th Conference on Robot Learning (CoRL 2024), Munich, Germany.

# 1   Introduction

A tactile memory system, which stores and retrieves tactile information, is essential for daily activities [1]. This study explores a robotic system, **Ta**ctile **Me**mory with **So**ft Ro**bot** (**TaMeSo-bot**), that integrates two key elements: soft robots, which enable safe physical interactions, and a tactile memory capable of storing and retrieving tactile information within a database for robust and fine contact-rich manipulation. Retrieval-based approaches eliminate the need for such labor-intensive segmentation and labeling processes on the database, and therefore have shown great potential in robot manipulation [2, 3, 4]. Nevertheless, building our system upon these methods presents a major challenge: The development of effective and robust representations capable of capturing subtask-relevant information in contact-rich scenarios is still an open problem.

This study proposes a Transformer-based tactile representation designed for retrieval-based manipulation in contact-rich tasks. We employ a distributed tactile sensor integrated into the gripper and a built-in force-torque (F/T) sensor on the robot's wrist. Sequences of tokens derived from each taxel and the other modalities—including force/torque signals, soft wrist poses, and end-effector poses—are provided as input to the transformer model. To obtain a robust feature representation relevant to the task, we train our model leveraging masked token prediction techniques [5, 6]. As shown in Fig. 1, these learned representations are stored in the tactile memory and subsequently used via retrieve-and-replay execution. During the online execution, we leverage the physical softness of the wrist to smooth the retrieved actions, which we refer to as *mechanical smoothing*.

We validated our method through offline and real-world robot experiments with peg-in-hole tasks. In offline evaluations, our model achieved more accurate action retrieval than a baseline without spatio-temporal interactions and masking. Also, the learned features were clearly segmented according to subtasks. In real-world experiments, the robot completed peg-in-hole tasks even under grasp and hole pose uncertainty, and with previously unseen peg sizes.

# 2   TaMeSo-bot: Tactile Memory with Soft Wrist

## 2.1   Tactile Memory System: Representation Learning

The tactile memory system implements a non-parametric control policy that retrieves the appropriate action by querying a database whose keys are tactile representations augmented with multimodal contexts. To build these query keys, we aim to learn an encoder $\mathcal{E}$ that maps a sub-trajectory $\tau = (\{s_{t-H+1}, a_{t-H+1}\}, \ldots, \{s_t, a_t\})$ consisting of state-action pairs $\{s, a\}$ over a history window $H$, to a compact representation vector $z_t$ that is later stored in a database with the corresponding action $a_t$. Each embedding $z_t$ is stored in the database together with its accompanying action $a_t$ and later used to retrieve the appropriate action at execution time.

**Tokenization and Position Encoding**   As shown on the left of Fig. 1, at each timestep $t$, we collect tactile signals from a PapillArray Tactile Sensor [7] with nine ($3 \times 3$) taxels, each producing 3D force measurements $s_t^{\text{tac}} = \{s_t^1, \ldots, s_t^9\} \in \mathbb{R}^{3 \times 9}$. As auxiliary states, we capture the wrist force/torque $u_t^{\text{ft}} \in \mathbb{R}^6$, arm pose $u_t^{\text{arm}}$, and gripper pose $u_t^{\text{grip}} \in \mathbb{R}^7$ measured from a motion capture system. We also record the robot action $a_t \in \mathbb{R}^6$, which is the displacement of the 3D position $[\Delta x, \Delta y, \Delta z]$ and 3D euler rotation representations $[\Delta \theta_x, \Delta \theta_y, \Delta \theta_z]$ of the robot arm.

The individual taxels and actions serve as **base tokens** $\{s^{\text{tac1}}, \ldots, s^{tac9}, a\}$, while the other sensory information $\{u^{\text{ft}}, u^{\text{arm}}, u^{\text{grip}}\}$ serves as auxiliary tokens that compensate for global contexts. These tokens are projected to the $d_e$-dimensional embeddings $b = \{b^1, \ldots, b^9, b^{10}\}$ and $e = \{e^{\text{ft}}, e^{\text{arm}}, e^{\text{grip}}\}$ by a linear projection $f(\cdot)$. To incorporate the spatial relationship of tactile tokens, each taxel position is encoded as a spatial position embedding $e^{\text{pos}} \in \mathbb{R}^{d_{\text{pos}}}$ by sinusoidal encoding, reflecting its location in the $3 \times 3$ grid. The action token is considered at the center of this grid to facilitate spatial reasoning between actions and tactile feedback. To incorporate these auxiliary embeddings as global context, we add their weighted sum to each base token embedding.

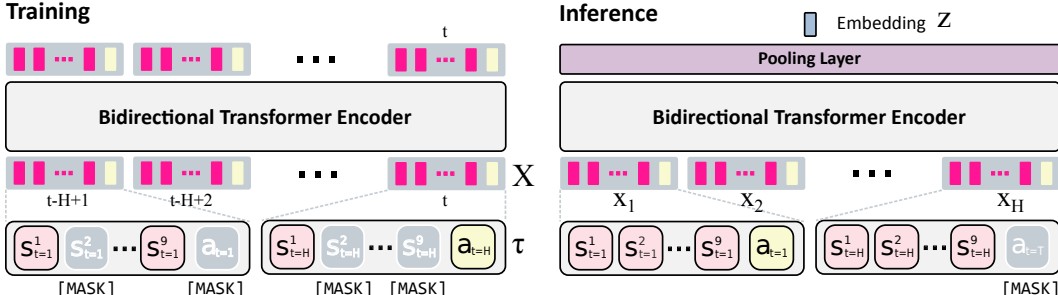

Figure 2: **Masked Trajectory Encoder** for distributed tactile sensors and actions. During training, the encoder learns to reconstruct the states and actions within a time window $H$, while randomly masking input tokens. After training, the tactile trajectory datasets are encoded into a single representative embedding $\mathbf{z}$ that captures the spatio-temporal dynamics of the tactile-action sequences.

In summary, the one token embeddings $\boldsymbol{x}^i \in \mathbb{R}^{d_e + d_{\mathrm{pos}}}$ are described as

$$\boldsymbol{x}^i = \left(\boldsymbol{b}^i + w^{\mathrm{ft}} \cdot \boldsymbol{e}^{\mathrm{ft}} + w^{\mathrm{arm}} \cdot \boldsymbol{e}^{\mathrm{arm}} + w^{\mathrm{grip}} \cdot \boldsymbol{e}^{\mathrm{grip}}\right) \oplus \boldsymbol{e}^{\mathrm{pos}}, \tag{1}$$

where $\oplus$ denotes hard-concatenation of the positional embeddings and $+$ denotes soft-concatenation with weights $w$ for each token embeddings.

**Bidirectional Transformer-based Tactile Encoder** Our model processes sub-trajectory sequences $\tau$ from historical observations. Through the embedding process defined in Eq. (1), a sub-trajectory $\tau$ is transformed into an input tensor $\boldsymbol{X} = \left(\{\boldsymbol{x}_1^1, \ldots, \boldsymbol{x}_1^{10}\}, \ldots, \{\boldsymbol{x}_H^1, \ldots, \boldsymbol{x}_H^{10}\}\right) \in \mathbb{R}^{H \times 10 \times d}$, where $H$ is the sequence length, 10 represents the number of tokens per timestep consisting of nine taxel tokens and one action token, and $d = d_e + d_{\mathrm{pos}}$ is the embedding dimension. We employ a bidirectional transformer encoder $\mathcal{E}$ with $L$ layers to process the input tensor $\boldsymbol{X}$. Details of the model parameters are provided in the Appendix.

**Masked Trajectory Encoder** To learn robust tactile representations that capture the intrinsic structure of the data, we employ masked token prediction strategy [5, 6]. As illustrated in Fig. 2, we train a Bidirectional Transformer-based encoder that learns to reconstruct input states and actions with a temporal window $H$ while randomly masking input tokens. Specifically, for each input $\boldsymbol{X}$, we sample a masking ratio uniformly from the range $[0, 0.6]$, following prior work [5]. During inference, since the current action $\boldsymbol{a}_t$ is unknown, we mask the action token in the current timestep and use the encoder to generate query vectors that capture the current tactile context.

**Feature Pooling** To obtain a tactile representation that captures information across all tokens, we apply a pooling function $g(\cdot)$. For our system, we adopt the widely used average pooling [8].

## 2.2 Tactile Memory System: Database and Retrieval

**Database Construction** Each trajectory $\tau$ consists of a sequence of tactile tokens $\mathbf{x}_t \in \mathbb{R}^{10 \times H \times d}$, where 10 denotes the number of tokens including nine tactile sensor taxels and one action token, $H$ is the number of timesteps to encode, and $d$ is the feature dimension. The pooling operation converts the input tensor $\mathbf{x}_t$ into a representative vector $\mathbf{z}_t \in \mathbb{R}^d$. The resulting pair $\{\mathbf{z}_t, \mathbf{a}_t\}$, where $\mathbf{a}_t$ is the corresponding action sequence, is then stored in the database $\mathcal{D}$.

**Non-parametric Control Policy** At execution time, given a sequence of tactile observation $(\boldsymbol{s}_{t-H+1}, \ldots, \boldsymbol{s}_t)$ and previous actions $(\boldsymbol{a}_{t-H+1}, \ldots, \boldsymbol{a}_{t-1})$, we encode it using the pretrained encoder $\mathcal{E}$ and pooling function $g(\cdot)$ to obtain the query representation $\mathbf{z}_q \in \mathbb{R}^d_e$. Here we mask the current action token as it is unknown at the time of execution. We then retrieve the $k$ most similar representations from the database using L2 distance, $\mathrm{dist}(\boldsymbol{z}_q, \boldsymbol{z}_i) = \|\boldsymbol{z}_q - \boldsymbol{z}_i\|$. For real-time robot control at 50 Hz, we employ an approximate nearest neighbor search using a hierarchical navigable small world (HNSW) graph-based index [9].

## 3 Experiments

We conducted a series of experiments to evaluate the robustness of our TaMeSo-bot system in peg-in-hole tasks. Our experiments are designed to validate the effectiveness of our proposed **TaMeSo-bot** system, aiming to answer a key question: Does the proposed masking strategy improve the quality and robustness of the learned tactile representation? To address these questions, we conduct comprehensive evaluations through both offline analysis using collected datasets and experiments on a real robotic system. The offline evaluation results are introduced in the Appendix.

**Robot Setup** We use a robotic arm (UR5e, Universal Robots A/S, Denmark) with a force-torque sensor, a parallel gripper (Hand-e, Robotiq, Canada), and a soft wrist [10], which consists of three springs and allows 6 degrees of freedom deformation. A motion tracker (HTC VIVE Tracker [11], HTC Corporation, Taiwan) is mounted on the gripper to capture its pose, and a tactile sensor (PapillArray Tactile Sensor [7], Contactile Pty Ltd, Australia) is attached to one side of the gripper.

**Tasks and Dataset Collection** We focus on peg-in-hole tasks. For the experiments, circular metal pegs with diameters of 15 mm and 20 mm are used. The tolerance between the pegs and holes is 1 mm. We collect demonstrations using the teleoperation system with a VR controller proposed in [12]. The robot grasps each peg in a vertical orientation. Both the sampling and control frequencies are set to 50 Hz. Each demonstration begins from a different initial pose and terminates when the peg is fully inserted into the hole, lasting approximately 150–300 time steps (3-6 seconds). The resulting dataset comprises 300 successful demonstrations: 150 with a 15 mm diameter peg and 150 with a 20 mm diameter peg.

**Task Setup** We evaluate task success rates under both seen and unseen conditions. We retrieve the actions of the $k = 3$ nearest neighbors and select one of them uniformly at random to prevent the robot from getting stuck during the trials. For the seen conditions, we compare our method with and without masking, using pegs with 15 mm and 20 mm diameters. The robot initiates each trial from four directions. For unseen conditions, we evaluate only the masked version of our method, using pegs with 10 mm and 25 mm diameters, and a 15 mm peg with a 10° grasp misalignment and a 3° hole tilt. The detail is introduced in the Appendix.

**Results** Table 1 reports the success rates from five trials at each of four initial positions, showing that the masked encoder consistently outperforms the unmasked baseline. Finally, Table 3 shows that the masked method maintains a success rate of roughly 50% on unseen tasks.

Table 1: **Real-robot evaluation in seen conditions.** Peg-in-hole task success rate (%) under four different starting positions.

| Method | 15 mm peg | 20 mm peg |
|---|---|---|
| Ours (w/o mask) | 50% (10/20) | 75% (15/20) |
| **Ours (full)** | **75% (15/20)** | **80% (16/20)** |

Table 2: Real-robot evaluation in unseen conditions: unseen diameters of pegs and different peg and hole poses.

Table 3: **Real-robot evaluation with unseen hole diameters and conditions** testing different peg sizes and pose variations.

| Diameters | Conditions | Success rate ↑ |
|---|---|---|
| 25 mm | – | 80% (8/10) |
| 10 mm | – | 50% (5/10) |
| 15 mm | Hole tilted 3° | 50% (5/10) |
| 15 mm | Peg misalignment 10° | 50% (5/10) |

## 4 Conclusion

We presented **TaMeSo-bot**, integrating tactile memory with soft robotics for robust contact-rich manipulation under uncertainty. Our transformer-based spatiotemporal tactile representation with masking successfully extracted task-relevant features from multiple sensor modalities. Experiments with peg-in-hole tasks validated our approach's effectiveness under grasp and hole pose uncertainty, even with previously unseen peg sizes.

# Appendix

## A   Attached File

Please see the supplementary video submitted. The video includes highlights of our proposed method, visualizations of embeddings throughout manipulation episodes, successful insertions with various peg-hole configurations, and failure cases that complement the results described in this appendix.

## B   Additional Detail for Proposed Method

### B.1   Learning Objective

After encoding, each processed token is passed through a modality-specific linear layer that projects the embedding back to the original signal dimension. The reconstructed outputs are then compared with the raw observations at the same temporal indices. The model is trained to reconstruct these observations by minimizing a mean-squared-error (MSE) loss averaged over every element in the entire mini-batch $U$:

$$\mathcal{L} = \mathcal{L}_{\text{tactile}} + \mathcal{L}_{\text{action}}, \tag{2}$$

where

$$\mathcal{L}_{\text{tactile}} = \frac{1}{|U|} \sum_{t \in U} \text{MSE}(\hat{\boldsymbol{s}}_t^{\text{tac}}, \boldsymbol{s}_t^{\text{tac}}), \qquad \mathcal{L}_{\text{action}} = \frac{1}{|U|} \sum_{t \in U} \text{MSE}(\hat{\boldsymbol{a}}_t, \boldsymbol{a}_t).$$

Here $\boldsymbol{s}_t^{\text{tac}} \in \mathbb{R}^{3 \times 9}$ stacks the nine 3-D taxel readings at time step $t$, $\text{MSE}(\cdot, \cdot)$ denotes the element-wise mean-squared error, and $|U|$ is the mini-batch size. The loss in Eq. (2) is optimized with the Adam optimizer. Details of the training parameters are provided in the Appendix.

### B.2   Selection of Hyperparameters

Table 4 shows our Bidirectional Transformer model hyperparameters. We used 248-dimensional token embeddings ($d_e$) with 8-dimensional positional embeddings ($d_{pos}$), yielding a 256-dimensional input embedding. The model uses 4 encoder layers, 8 attention heads, a feed-forward size of 512, and 0.1 dropout. These parameters were selected based on preliminary experiments, where we systematically evaluated different configurations. Specifically, we tested various embedding and hidden size pairs from $\{(32, 64), (64, 128), (128, 256), (256, 512), (512, 1024)\}$, with $(256, 512)$ yielding the best action retrieval accuracy. To facilitate reproduction of our results, the lower section of the table summarizes the training hyperparameters used in our experiments.

### B.3   Mechanical Smoothing

Encoding a history window already promotes temporal consistency in the retrieved actions, yet small discontinuities can still cause the robot to overshoot the hole or collide with the surface, risking damage to the environment. This risk is particularly significant in industrial settings where rigid, position-controlled robots are used. To address these discontinuities, we introduce mechanical smoothing that exploits the soft wrist's compliance to smooth the commanded motion.

The soft wrist in our system serves as a compliant element. However, it also exhibits a delayed response, similar to other soft robots [13]. While such delays are typically compensated for through specific control techniques [13], we instead leverage this delayed response as a low-pass filtering property. The soft wrist naturally filters high-frequency components of commanded actions, providing mechanical low-pass filtering that reduces the impact of discontinuities inherent in the retrieval-based policy. The physical compliance of the soft wrist complements our mechanical smoothing

Table 4: Hyperparameters of our method.

| Hyperparameter | Notation | Value |
|---|---|---|
| Dimension of token embeddings | $d_e$ | 248 |
| Dimension of positional embeddings | $d_{\text{pos}}$ | 8 |
| Input embedding size | $d = d_e + d_{\text{pos}}$ | 256 |
| Temporal window size | $H$ | 15 |
| Number of attention heads | $N_{\text{head}}$ | 8 |
| Feed-forward hidden dimension | $N_{\text{hid}}$ | 512 |
| Number of encoder layers | $N_{\text{layers}}$ | 4 |
| Dropout | | 0.1 |
| Epochs | | 30 |
| Learning rate | | $1.0 \times 10^{-4}$ |
| Batch size | | 128 |
| # Data-loading workers | | 16 |
| Train / validation split ratio | | 0.8 |
| Maximum masking ratio of input tokens | | 0.6 |

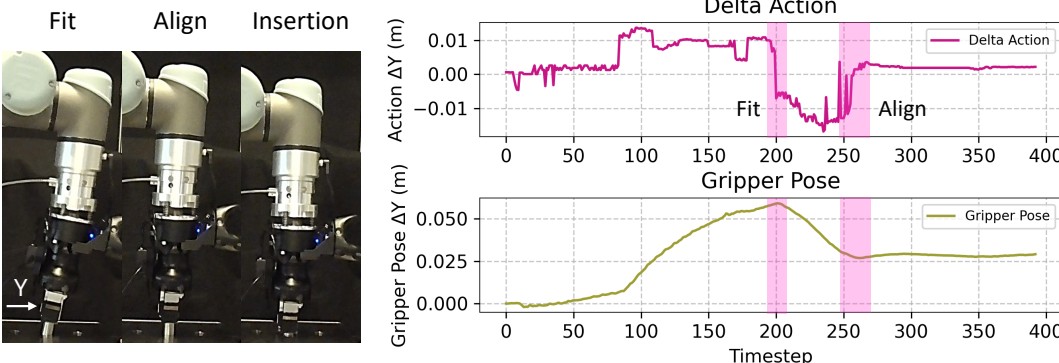

Figure 3: *Left*: Snapshots of three phases during an insertion trial. *Right*: Time transitions of the commanded $\Delta y$ action (top) and the resulting gripper $y$-position (bottom). The shaded regions indicate abrupt command changes.

by naturally absorbing discontinuities in the commanded actions. Figure 3 illustrates a representative successful trial with snapshots and time-series plots of the retrieved action commands and the gripper pose. Despite abrupt command changes (shaded in pink), the gripper motion remains smooth.

## C  Offline Evaluation

### C.1  Evaluation of Action Retrieval

We evaluate the quality of our learned spatiotemporal tactile representation by assessing its performance as a retriever policy. We construct a database $\mathcal{D}$ of paired tactile representations and actions $\{z, a\}$, where a query tactile embedding $z_q$ retrieves the most similar representation $z$ and returns its associated action $a$. For this evaluation, we randomly allocate 80% of the collected sequences to build the database and use the remaining 20% as test queries.

**Metrics**  For each tactile state in the test set, we retrieve the nearest neighbor ($k = 1$) from the database and measure the Root Mean-Squared Errors (RMSE), defined as the element-wise L2 distance between the retrieved action and the ground-truth action executed at that time step. Lower

Table 5: **Offline Evaluation.** Action prediction accuracy using nearest neighbor retrieval based on learned representations. Metrics include RMSE (with standard deviation), median (p50), and 95th percentile (p95) error values. Lower values indicate better prediction accuracy across all metrics.

| Method | Δ Position Errors [mm] | | | Δ Rotation Errors [°] | | |
| --- | --- | --- | --- | --- | --- | --- |
| | RMSE | p50 | p95 | RMSE | p50 | p95 |
| Naive State-Action Concatenation | $8.055 \pm 6.776$ | 10.975 | 24.648 | $0.837 \pm 0.442$ | 0.612 | 1.622 |
| Ours (w/o Masking) | $6.536 \pm 6.036$ | 8.312 | 21.506 | $0.644 \pm 0.331$ | 0.485 | 1.190 |
| **Ours (Full)** | **$6.025 \pm 5.456$** | **7.803** | **19.732** | **$0.629 \pm 0.309$** | **0.486** | **1.109** |

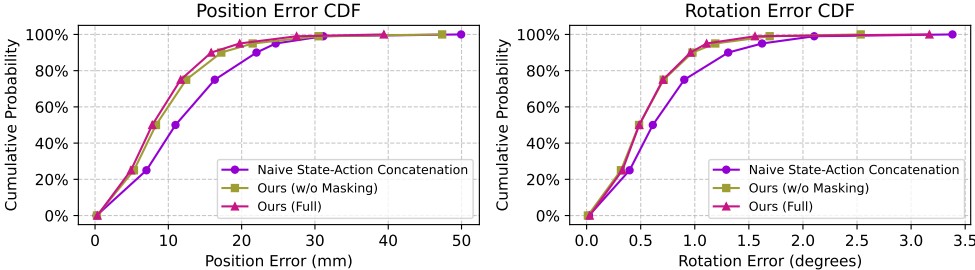

Figure 4: Cumulative distribution of position errors (left) and rotation errors (right) for Naive State-Action Concatenation (●), Ours without masking (■), and Ours with full masking (▲).

MSE values indicate more accurate action retrieval, effectively evaluating how well our representation learning enables the nearest-neighbor policy to retrieve appropriate actions.

**Baselines** Since no existing methods directly address the joint representation learning of distributed tactile signals and actions in our problem setting, we compare our approach against the following baselines to demonstrate the effectiveness of our proposed method:

- **Naive State-Action Concatenation**: A baseline that hard-concatenates non-encoded sensor data with actions, following the state-action fusion strategy adopted in [4], representing a simple retrieval approach without learned encoding.

- **Ours (w/o Masking)**: A variant of our proposed Transformer architecture that learns joint tactile-action representations but without implementing the masking strategy, demonstrating the contribution of our basic representation learning approach.

**Results** Table 5 summarizes the results for the offline evaluation of the retrieval policy performance on collected dataset. Our full method drastically reduces the position RMSE and rotation RMSE compared to naive concatenation. The improvement is particularly evident in the median (p50) position error, and in the 95th percentile (p95) position error. This indicates that our method not only improves average performance but also significantly enhances robustness in challenging cases. Figure 4 illustrates the superior performance of our proposed representation compared to the naive feature concatenation baseline. The cumulative distribution plots of both position and rotation errors demonstrate that our approach consistently achieves lower error rates across all percentiles. The ablation study further confirms the effectiveness of our masking strategy. Comparing "Ours (w/o Masking)" to "Ours (Full)", we observe that incorporating the masking mechanism provides consistent improvements across error distributions. More notably, the 95th percentile errors demonstrate that masking particularly enhances performance on difficult samples. We also perform visualization for the latent space in the Appendix.

## C.2 Embedding Space Analysis

We analyze the structure of the learned high-dimensional embedding space to assess whether our representation captures semantically meaningful information corresponding to distinct subtasks within the contact-rich manipulation process. We create the database using 80% of the collected demonstrations and evaluate the distance to one query vector picked up from the remaining 20% trajectories. To visualize this high-dimensional structure while preserving relevant distance relationships, we employ a two-pronged approach. First, we use t-SNE to project the state embeddings from the offline dataset into a lower-dimensional (2D) space. Second, to illustrate how high-dimensional proximity relates to subtasks, we overlay colormaps onto the t-SNE projection. These colormaps represent the distances calculated in the original high-dimensional embedding space between each projected state and representative query embeddings selected for key subtasks: "*fit*", "*align*", and "*insert*". The representative query embeddings are selected from manually segmented subtask sequences, specifically choosing states positioned in the middle of each subtask sequence.

Figure 5 visualizes the learned embedding space of our proposed method (**top row**) and the variants of our method without masked token prediction (**bottom row**). The top row shows the embedding space of our proposed method, where we observe distinctive clusters for each subtask. The bottom row illustrates the embedding space of our variant without masked token prediction. In the bottom row without masked token prediction, we observe that compared to the results with masked token prediction, there are more embeddings with closer distances to the query points, resulting in more scattered clusters. Comparing both rows reveals that our proposed method produces more distinctive representations, which likely contributes to the improved performance in offline evaluation. Interestingly, we observe that "*fit*", "*align*", and "*insert*" are positioned near each other, indicating seamless transitions between these subtasks. "*Insert*" appears as a subset of "*align*", forming a smaller cluster, which aligns with the physical nature of these manipulation phases. States closer to a specific subtask query in the high-dimensional space tend to cluster together in the t-SNE projection, suggesting the learned representation is well-suited for retrieval-based control by capturing task progress. For visualization of distances to query points throughout an episode aligned with real-robot's videos, please refer to the enclosed video.

## D   More Ablation Studies

Due to page limitations in the main text, we provide more detailed ablation studies here to thoroughly examine the effectiveness of our approach.

### D.1   Effect of context weights $w^{\text{ft}}$, $w^{\text{arm}}$, $w^{\text{grip}}$

As described in **Section 2.1** of the main text, we augment our main tokens of distributed taxels by soft-concatenating inputs from various sensors as auxiliary tokens with appropriate weights to provide contextual information. To investigate the contribution of each context, we toggled the weights assigned to force/torque sensors ($w^{\text{ft}}$), arm pose ($w^{\text{arm}}$), and gripper pose ($w^{\text{grip}}$), and evaluated action retrieval performance in offline evaluation with and without these auxiliary contexts.

Table 6 shows that disabling all auxiliary contexts significantly degraded performance. Among these contextual inputs, force/torque information had the most substantial impact, with its inclusion markedly improving performance. The contributions of arm pose and gripper pose were nearly equivalent, though we observed that arm pose primarily influenced position accuracy, while gripper pose had a greater effect on rotation accuracy. Specifically, the model without gripper pose (w/o Gripper Pose) achieved the best position RMSE of 5.90 mm, while the model without arm pose (w/o Arm Pose) attained the best rotation RMSE of 0.623°.

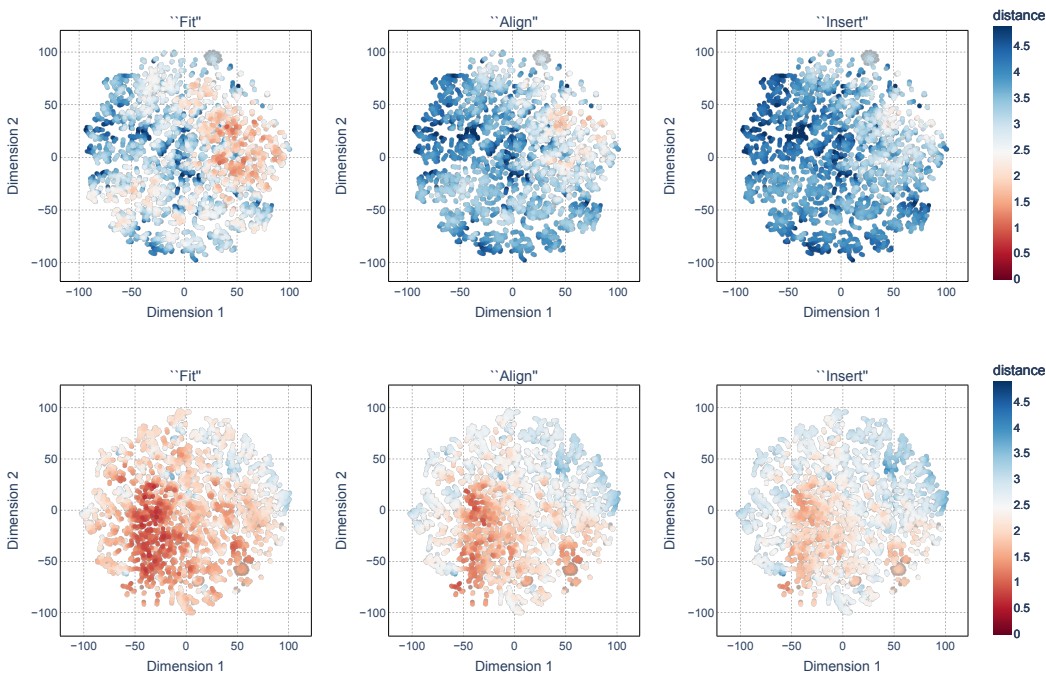

Figure 5: **t-SNE visualization of the learned embedding space**. Colormaps indicate distance in the original high-dimensional space to representative query points for each subtask. Top: Embedding space visualization of our method, showing distinctive distances to query points representing *fit*, *align*, and *insert* subtasks. Bottom: Visualization of the variant of our proposed method (without masked token prediction). Compared to the top row, distances to query points are less distinctive and more scattered. Query points for *fit* and *align*, and *align* and *insert* show proximity to each other, indicating their seamless task nature. The query point for *insert* shows distance patterns suggesting it functions as a subset of *align*, forming a more concentrated region of similarity.

Table 6: Position and Rotation Errors of action retrieval in offline evaluation for varying context weights. We trained models with different combinations of auxiliary contexts ($w^{\text{ft}}$ for force/torque, $w^{\text{arm}}$ for arm pose, and $w^{\text{grip}}$ for gripper pose) and evaluated their performance on offline trajectories. Lower values indicate better action retrieval performance.

| Input Contexts | $w^{\text{ft}}$ | $w^{\text{arm}}$ | $w^{\text{grip}}$ | Position RMSE [mm] | Rotation RMSE [°] |
|---|---|---|---|---|---|
| No Auxiliary Context | 0.0 | 0.0 | 0.0 | 7.53 | 0.722 |
| w/o Force/Torque | 0.0 | 0.4 | 0.4 | 6.29 | 0.647 |
| w/o Arm Pose | 0.2 | 0.0 | 0.4 | 5.95 | **0.623** |
| w/o Gripper Pose | 0.2 | 0.4 | 0.0 | **5.90** | 0.664 |

## D.2 Effect of the size of temporal window $H$

As described in **Section 3.1** of the main text, we input a history of state and action pairs to the Transformer. To investigate the best value for this temporal window size $H$, we conducted an ablation study comparing four variants ($H = 5, 10, 15, 20$). Table 7 summarizes the results. We observe a trade-off between position and rotation accuracy. Smaller windows yield slightly better positional performance at the cost of higher rotational error, while larger windows improve rotation at the expense of position. Based on this balance, we selected $H = 15$ for the experiments.

Table 7: **Temporal Window Size Comparison.** Position and rotation error statistics for different temporal window sizes. Metrics include RMSE (with standard deviation), median (p50) and 95th percentile (p95) error values.

| Temporal Window | Position Errors [mm] | | | Rotation Errors [°] | | |
|---|---|---|---|---|---|---|
| | RMSE | p50 | p95 | RMSE | p50 | p95 |
| 5 | $5.984 \pm 5.402$ | 7.673 | 19.479 | $0.672 \pm 0.348$ | 0.497 | 1.244 |
| 10 | $5.992 \pm 5.431$ | 7.673 | 19.576 | $0.635 \pm 0.318$ | 0.482 | 1.119 |
| 15 | $6.025 \pm 5.456$ | 7.803 | 19.732 | $0.629 \pm 0.309$ | 0.486 | 1.109 |
| 20 | $6.129 \pm 5.506$ | 7.946 | 19.568 | $0.614 \pm 0.306$ | 0.468 | 1.090 |

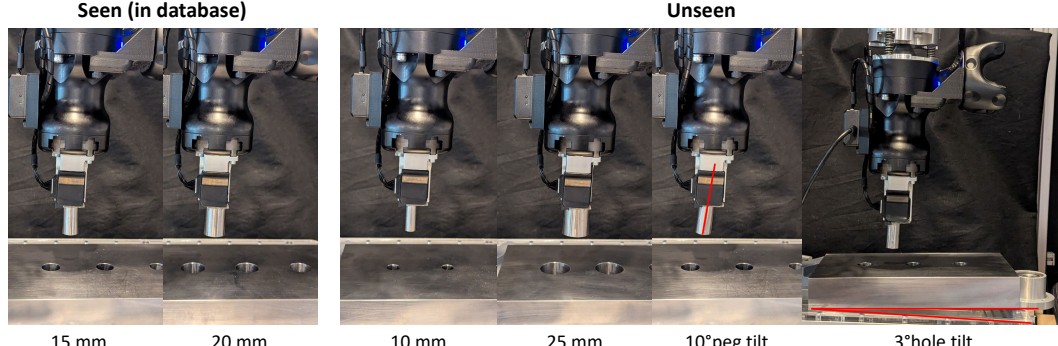

| Seen (in database) | | Unseen | | | |
|---|---|---|---|---|---|
| 15 mm | 20 mm | 10 mm | 25 mm | 10°peg tilt | 3°hole tilt |

Figure 6: Pegs and holes used in the experiments.

# E  Additional Experimental Details

We used the teleoperation system proposed in [12] for both data collection and retrieval execution. In this system, Virtual Reality (VR) controllers specify both the desired end-effector pose and the stiffness parameters fed into the underlying compliance controller [14]. Table 8 shows the compliance controller parameters. For any six-dimensional array, the first three elements correspond to the translational axes (X, Y, Z) and the last three to the rotational axes (roll, pitch, yaw). The stiffness parameters were kept constant throughout the experiments. The proportional gains determine how quickly the robot responds to deviations in each Cartesian axis, while the derivative gains provide damping to resist rapid changes. The error scale uniformly adjusts the magnitude of the computed P/D error signals to tune overall sensitivity. The number of internal iterations controls how many forward-dynamics simulation steps the controller takes per cycle to reconcile force and motion commands. Readers are referred to [14] for more details on the compliance controller.

We used a soft wrist [10], consisting of three coil springs. The spring constant is 4.112 N/mm, and the equilibrium length is 25 mm. Figure 6 shows the pegs and holes used in the experiments.

Table 8: Parameters of the compliance controller.

| Parameter | Value |
|---|---|
| Stiffness | [1200, 1200, 1200, 300, 300, 300] |
| Proportional gains | [0.035, 0.035, 0.035, 0.5, 0.5, 0.5] |
| Derivative gains | [0, 0, 0, 0, 0, 0] |
| Error scale | 0.8 |
| Iterations | 1.0 |

# F    Discussion on Real-World Experimental Results

## F.1    Effect of the Masked Token Prediction

Our real-robot experiments in **Section 4.2** demonstrate that masked token prediction enhances task performance. By randomly masking portions of the input sequence, the encoder is forced to infer missing information from both past and future context, yielding richer spatiotemporal feature representations. As a result, the three core phases of peg-in-hole ("*fit*", "*align*", and "*insert*") emerge as well-separated clusters in the learned embedding space, as visualized in Figure 5. This clearer clustering enables the nearest-neighbor retrieval to more reliably select appropriate actions for each phase, reducing erroneous transitions and improving overall success rates. In contrast, the unmasked baseline produces less distinctive embeddings, leading to less consistent action retrieval and lower insertion success. These findings confirm that masking not only produces more robust representation but also sharpens phase-specific distinctions, which improves the following action retrieval performance.

## F.2    Failure Cases

The most common failure occurred when multiple insertion attempts timed out after 30 seconds. This typically resulted from the retrieved action direction deviating slightly from the true hole direction. The problem was most pronounced with the unseen 10 mm peg and hole. The small hole makes it difficult for the robot to locate the hole before the time limit expires. We also observed that, during the search phase, the robot occasionally inserted the peg into an adjacent hole instead of the intended target. Interestingly, once the peg "*fits*" into the wrong hole, the system still transitions to the *align* phase, despite the hole's location being markedly different from the goal. This behavior suggests that successful "*fit*" embeddings lie close together in the embedding space, regardless of which hole is engaged. Another failure mode occurred when the embedding fell outside the distribution of the training demonstrations. For example, when inserting into a tilted hole, the embedding lies far from the data manifold defined by the demonstrations. As a result, the robot repeatedly retrieves the same nearest action, even though that action is a poor match, and ends up in an unnatural, stalled configuration. Concrete examples of these failures are shown in the supplemental video.

# G    Limitation

While our method shows promising results even under grasp and hole pose uncertainty and with unseen pegs, there are several limitations that should be addressed in future work. First, due to the extremely high cost of collecting real-robot demonstrations across multiple tasks and diverse configurations, we limited our study to peg-in-hole tasks. This limitation could potentially be addressed through ongoing efforts in the community to develop large-scale robot manipulation datasets [15, 16], which would enable the validation of our approach on a wider range of contact-rich manipulation tasks. Second, our approach assumes that the query and the embedding in the database are from reasonably close domains. For queries that fall completely outside the domain covered by the database, our method cannot extrapolate expected outputs. An interesting direction for future research would be to explore method for transferring embedding spaces between different domains, which could enable more flexible adaptation to novel manipulation scenarios without requiring extensive data collection. Lastly, this study uses an external motion tracker to capture the pose of the soft wrists. Future work could explore incorporating visual inputs or modeling the entire system as a partially observable Markov decision process (POMDP) to realize a tracker-less system. Such an approach would simplify the system setup and enable broader reproduction of our method across diverse scenarios.

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
