# OpenReview forum: "Tactile Memory with Soft Robot: Tactile Retrieval-based Contact-rich Manipulation with a Soft Wrist"
_robot-learning.org/CoRL/2025/Workshop/Dexterous_Manipulation — CoRL 2025 Workshop Dexterous Manipulation Spotlight_

### Official Review · Reviewer_bmx5 · 2025-09-08
**Tactile Memory with Soft Robot: Tactile Retrieval-based Contact-rich Manipulation with a Soft Wrist -Good summary of research work focused on retrieving data from touch-based examples. Good evidence based presentation**

**Rating:** 7
**Confidence:** 2

**Review:**

Tactile Memory with Soft Robot: Tactile Retrieval-based Contact-rich Manipulation with a Soft Wrist

Good summary of research work focused on retrieving data from touch-based examples. Good evidence based presentation.

Although not the most complex research topic for this area the work is well thought through and the evidence base well built up. Data is collected and used from multiple sources. The equipment and sensors chosen for the work are some of the best available for this task. This work then build a new movement and self correcting action. Manipulating the specific movement to complete the specific task

---

### Official Review · Reviewer_EHms · 2025-09-11
**Interesting method for retrieval-based manipulation**

**Rating:** 7
**Confidence:** 4

**Review:**

Strengths:

TaMeSo-bot presents a novel method for converting tactile and proprioceptive action data into a structured embedding space, used for known-action retrieval at test time. Such an approach I imagine can be helpful for noisy tactile data or other situations where standard methods fail. The authors show their features learned are a competitive method for tasks tested and outperform presented baselines.  Results seem reasonable in seen and unseen conditions. Figures are of high quality and help illustrate their approach.


Weaknesses:

Comparisons are limited to other retrieval-style baselines (naive concat, no-masking transformer). A comparison to diffusion policy, bc, or another common parametric baseline would strengthen the argument for non-parametric methods. One interesting direction of work could be to demonstrate that TaMeSo-bot is robust to noise in tactile data, as the method can still "snap" to a known safe action in the presence of noise (versus diffusion policy etc., which might output unsafe actions from OOD tactile data).

The method is only demonstrated on one suite of tasks (peg-in-hole), though several perturbations are tested.


Minor:

L60 typo spatilal→ spatial

---

### Decision · Program_Chairs · 2025-09-18

Accept (Spotlight)